# Revisiting Random Weight Perturbation for Efficiently Improving Generalization

**Tao Li**                                                    *li.tao@sjtu.edu.cn*
*Shanghai Jiao Tong University*

**Qinghua Tao**                                              *qtao@esat.kuleuven.be*
*KU Leuven*

**Weihao Yan**                                          *ywh926934426@sjtu.edu.cn*
*Shanghai Jiao Tong University*

**Yingwen Wu**                                          *yingwen_wu@sjtu.edu.cn*
*Shanghai Jiao Tong University*

**Zehao Lei**                                             *lzhsjstudy@sjtu.edu.cn*
*Shanghai Jiao Tong University*

**Kun Fang**                                             *fanghenshao@sjtu.edu.cn*
*Shanghai Jiao Tong University*

**Mingzhen He**                                        *mingzhen_he@sjtu.edu.cn*
*Shanghai Jiao Tong University*

**Xiaolin Huang**[*]                                    *xiaolinhuang@sjtu.edu.cn*
*Shanghai Jiao Tong University*

**Reviewed on OpenReview:** *https://openreview.net/forum?id=WbbgOHpoPX*

## Abstract

Improving the generalization ability of modern deep neural networks (DNNs) is a fundamental challenge in machine learning. Two branches of methods have been proposed to seek flat minima and improve generalization: one led by sharpness-aware minimization (SAM) minimizes the worst-case neighborhood loss through adversarial weight perturbation (AWP), and the other minimizes the expected Bayes objective with random weight perturbation (RWP). While RWP offers advantages in computation and is closely linked to AWP on a mathematical basis, its empirical performance has consistently lagged behind that of AWP. In this paper, we revisit the use of RWP for improving generalization and propose improvements from two perspectives: i) the trade-off between generalization and convergence and ii) the random perturbation generation. Through extensive experimental evaluations, we demonstrate that our enhanced RWP methods achieve greater efficiency in enhancing generalization, particularly in large-scale problems, while also offering comparable or even superior performance to SAM. The code is released at `https://github.com/nblt/mARWP`.

## 1 Introduction

Modern deep neural networks (DNNs) are commonly characterized by their over-parameterization, boasting millions or even billions of parameters. This extensive model capacity empowers DNNs to explore a vast

---

[*]Corresponding author

hypothesis space and achieve state-of-the-art performance across various domains (Tan & Le, 2019; Kolesnikov et al., 2020; Liu et al., 2021; Radford et al., 2021). However, this abundance of parameters relative to the number of training samples makes DNNs prone to memorizing the training data, leading to overfitting issues. Consequently, it becomes crucial to develop effective training algorithms that facilitate generalization beyond the training set (Neyshabur et al., 2017).

Numerous studies have been dedicated to improving the generalization ability of DNNs (Szegedy et al., 2016; Izmailov et al., 2018; Zhang et al., 2018; 2019; Foret et al., 2020). Building upon the notion that flat minima are better suited to adapt to potential distribution shifts between training and test data, leading to improved generalization (Hochreiter & Schmidhuber, 1997; Dinh et al., 2017; Li et al., 2018), two prominent branches of methods have emerged to identify and exploit such flat minima for effective generalization improvement. The first branch formulates the optimization objective as a min-max problem, aiming to minimize the training loss under the worst-case adversarial weight perturbation (AWP). This approach, also known as sharpness-aware minimization (SAM) (Foret et al., 2020), seeks to find flat minima that reside in neighborhoods characterized by consistently low loss values. The second branch, exemplified by LPF-SGD (Bisla et al., 2022), aims to recover flat minima by minimizing the expected training loss through random weight perturbation (RWP). Notably, these two approaches share similarities in their formulations and can be mathematically connected (Möllenhoff & Khan, 2023).

Despite achieving state-of-the-art generalization performance, AWP, represented by SAM (Foret et al., 2020), suffers from a significant drawback in terms of computation and training time. This is due to the involvement of two gradient steps, which doubles the computational requirements. Consequently, applying AWP to large-scale problems becomes a prohibitive challenge. On the other hand, RWP offers a more computationally efficient alternative, requiring only negligible additional computational overhead compared to regular training. However, it is commonly believed that RWP exhibits inferior empirical performance when compared to AWP (Zheng et al., 2021; Liu et al., 2022b). This discrepancy can be attributed to the fact that RWP perturbs the model with less intensity than AWP, which benefits from leveraging precise gradient information.

In this paper, we revisit the use of RWP for improving generalization and aim to bridge the performance gap between these two types of perturbations. We start by illustrating a trade-off between generalization and convergence in RWP: it requires perturbations with orders of magnitude larger than those needed in AWP, to effectively enhance generalization; however, this can lead to convergence issues in RWP. To tackle this challenge, we propose a simple approach called mixed-RWP, or m-RWP, which incorporates the gradient of the original loss objective to improve convergence and simultaneously guide the network towards better minima. Notably, although both SAM and m-RWP require two gradient steps per iteration, m-RWP is more efficient in improving generalization, which lies in two aspects: 1) in m-RWP, these two steps are separable and can be efficiently computed *in parallel*, enabling the same training speed as regular SGD. In contrast, the two gradient steps in SAM are successive, resulting in a doubling of the training time. 2) the two separable gradient steps in m-RWP allow simultaneous use *two different batches* of data, further accelerating the convergence, especially on large-scale datasets. In contrast, SAM does not allow for using two different batches and may even negatively impact generalization performance. The improved convergence of m-RWP also allows for a larger perturbation variance to confer a better trade-off between generalization performance and convergence. Furthermore, we improve the generation of random weight perturbation by incorporating historical gradient information as guidance. This improvement enables more stable and adaptive weight perturbation generation, leading to enhanced performance. As a result, we significantly boost the generalization performance of RWP and introduce two improved RWP approaches: 1) ARWP which achieves competitive performance to AWP but requires only half of the computation and 2) m-ARWP which achieves comparable or even superior performance while benefiting from parallel computing of the two gradient steps.

In summary, we make the following contributions:

- We analyze the convergence properties of SGD with RWP in non-convex settings and identify a potential trade-off between generalization and convergence in RWP. We then propose a simple method to enhance such trade-off and improve the generalization performance of RWP.

- We propose an improvement to the generation of random weight perturbation by utilizing historical gradient information. This enhancement enables more stable and adaptive generation of weight perturbations, leading to improved performance.

- We present a comprehensive empirical study showing that our improved RWP approaches can achieve more efficient generalization improvements compared to AWP, especially on large-scale problems.

## 2 Related Work

**Flat Minima and Generalization.** The connection between the flatness of local minima and generalization has been extensively studied (Dinh et al., 2017; Keskar et al., 2017; Izmailov et al., 2018; Li et al., 2018; Jiang* et al., 2020). Hochreiter *et al.* (Hochreiter & Schmidhuber, 1994; 1997) are among the first to reveal the connection between flat minima and the generalization of a model. Keskar *et al.* (Keskar et al., 2017) observe that the performance degradation of large batch training is caused by converging to sharp minima. More recently, Jiang *et al.* (Jiang* et al., 2020) present a large-scale study of generalization in DNNs and demonstrate a strong connection between the sharpness and generalization error under various settings and hyper-parameters. Keskar *et al.* (Keskar et al., 2017) and Dinh *et al.* (Dinh et al., 2017) state that the flatness can be characterized by Hessian's eigenvalues and provide computationally feasible method to measure it.

**Sharpness-aware Minimization (SAM).** SAM (Foret et al., 2020) is a recently proposed training scheme that seeks flat minima by formulating a min-max problem and utilizing adversarial weight perturbation (AWP) to encourage parameters to sit in neighborhoods with uniformly low loss. It has shown power to achieve state-of-the-art performance. Later, a line of works improves the SAM's performance from the perspective of the neighborhood's geometric measure (Kwon et al., 2021; Kim et al., 2022; Liu et al., 2022b) or surrogate loss function (Zhuang et al., 2022). Several methods have been developed to improve training efficiency (Du et al., 2022a;b; Liu et al., 2022a; Mi et al., 2022; Zhao et al., 2022b;b; Jiang et al., 2023; Li et al., 2024).

**Random Weight Perturbation (RWP).** RWP is widely used in deep learning. Multiple weight noise injection methods have been shown to effectively escape spurious local optimum (Zhou et al., 2019) and saddle points (Jin et al., 2021). Upon generalization, Zhang *et al.* (Zheng et al., 2021) discuss that RWP is much less effective for generalization improvement than AWP. Wen *et al.* (Wen et al., 2018) propose SmoothOut framework to smooth out the sharp minima. Wang & Mao (2021) propose Gaussian model perturbation (GMP) as a regularization scheme for SGD training, but it remains inefficient due to the need of multiple computation budgets for noise sampling. Bisla *et al.* (Bisla et al., 2022) connect the smoothness of the loss objective to generalization and adopt filter-wise random Gaussian perturbation generation to improve the performance. However, the performance of RWP still lags behind that of AWP (Bisla et al., 2022; Liu et al., 2022b). Notably, recent Möllenhoff *et al.* (Möllenhoff & Khan, 2023) mathematically connect the expected Bayes loss under RWP with the min-max loss in SAM and suggest that RWP can be viewed as a 'softer' version of AWP. We significantly lift the performance of RWP from the convergence perspective and fill the performance gap to that of AWP.

## 3 Preliminary

Let $f(\boldsymbol{x}; \boldsymbol{w})$ be the neural network function with trainable parameters $\boldsymbol{w} \in \mathbb{R}^d$, where $d$ is the number of parameters. The loss function over a pair of data point $(\boldsymbol{x}_i, \boldsymbol{y}_i)$ is denoted as $L(f(\boldsymbol{x}_i; \boldsymbol{w}), \boldsymbol{y}_i)$ (shorted for $L_i(\boldsymbol{w})$). Given the datasets $\mathcal{S} = \{(\boldsymbol{x}_i, \boldsymbol{y}_i)\}_{i=1}^n$ drawn from data distribution $\mathcal{D}$ with i.i.d. condition, the empirical loss can be defined as $L(\boldsymbol{w}) = \frac{1}{n} \sum_{i=1}^n L_i(\boldsymbol{w})$.

Two branches of methods are proposed to pursue flat minima and better generalization ability. The first, known as sharpness-aware minimization (Foret et al., 2020), tries to minimize the worst-case loss in a neighborhood (defined by a norm ball) to bias training trajectories towards flat minima, i.e.,

$$L^{\text{SAM}}(\boldsymbol{w}) = \max_{\|\boldsymbol{\epsilon}_s\|_2 \leq \rho} L(\boldsymbol{w} + \boldsymbol{\epsilon}_s), \tag{1}$$

where $\rho$ is the radius that controls the neighborhood size. Instead of posing the strict 'max-loss' over the neighborhood, the second, represented by LPF-SGD (Bisla et al., 2022), adopts 'expected-loss' and minimizes the posterior (typically an isotropic Gaussian distribution) of the following Bayes objective (Möllenhoff & Khan, 2023):

$$L^{\text{Bayes}}(\boldsymbol{w}) = \mathbb{E}_{\boldsymbol{\epsilon}_r \sim \mathcal{N}(\mathbf{0}, \sigma^2 \boldsymbol{I})} L(\boldsymbol{w} + \boldsymbol{\epsilon}_r), \tag{2}$$

where $\sigma^2$ is the variance that governs the magnitude of the random weight perturbation. Such expected loss objective can effectively smooth the loss landscape and thereby recover flat minima (Bisla et al., 2022).

The above two objectives resemble in formulation except that the maximum in Eqn. (1) is replaced by an expectation in Eqn. (2). Intuitively, the expectation could be viewed as a 'softer' version of the maximum. Möllenhoff & Khan (2023) demonstrates that the above two objectives can be bridged mathematically, leveraging the tools of Fenchel biconjugate in convex optimization (Hiriart-Urruty & Lemaréchal, 2004). Specifically, let $L^{\text{relaxed}}(\boldsymbol{w})$ be the Fenchel biconjugate of $L^{\text{Bayes}}(\boldsymbol{w})$ defined in the dual spaces of exponential-family distributions (Wainwright et al., 2008), which is a convex relaxation w.r.t. original Bayes objective. There exists an equivalence such that

$$\arg\min_{\boldsymbol{w}} L^{\text{SAM}}(\boldsymbol{w}) = \arg\min_{\boldsymbol{w}} L^{\text{relaxed}}(\boldsymbol{w}), \tag{3}$$

assuming the SAM-perturbation satisfies $\|\boldsymbol{\epsilon}_s\| = \rho$ at a stationary point.

**Assumptions.** Before delving into our analysis, we first make some standard assumptions in stochastic optimization which are typical as in Duchi et al. (2012); Ghadimi & Lan (2013); Karimi et al. (2016); Andriushchenko & Flammarion (2022); Jiang et al. (2023) that will be used in our theoretical analysis.

**Assumption 1 (Bounded variance)** *There exists a constant $M > 0$ for any data batch $\mathcal{B}$ such that*

$$\mathbb{E}\left[\|\nabla L_{\mathcal{B}}(\boldsymbol{w}) - \nabla L(\boldsymbol{w})\|_2^2\right] \leq M, \quad \forall \boldsymbol{w} \in \mathbb{R}^d. \tag{4}$$

**Assumption 2 ($\alpha$-Lipschitz continuity)** *Assume the loss function $L : \mathbb{R}^d \mapsto \mathbb{R}$ to be $\alpha$-Lipschitz continuous. There exists $\alpha > 0$ such that*

$$\|L(\boldsymbol{w}) - L(\boldsymbol{v})\|_2 \leq \alpha \|\boldsymbol{w} - \boldsymbol{v}\|_2, \quad \forall \boldsymbol{w}, \boldsymbol{v} \in \mathbb{R}^d. \tag{5}$$

**Assumption 3 ($\beta$-smoothness)** *Assume the loss function $L : \mathbb{R}^d \mapsto \mathbb{R}$ to be $\beta$-smooth. There exists $\beta > 0$ such that*

$$\|\nabla L(\boldsymbol{w}) - \nabla L(\boldsymbol{v})\|_2 \leq \beta \|\boldsymbol{w} - \boldsymbol{v}\|_2, \quad \forall \boldsymbol{w}, \boldsymbol{v} \in \mathbb{R}^d. \tag{6}$$

## 4 Random Weight Perturbation v.s. Adversarial Weight Perturbation

Despite the theoretical connection established between the 'max-loss' and 'expected-loss' objectives, the performance of the latter still empirically lags behind the performance of the former, as the biconjugate function could not be attained in real neural network functions and there are gaps between the original objectives and its approximation. In this section, we work on a practical analysis for solving the above two objectives through the lens of weight perturbation.

**Adversarial Weight Perturbation (AWP).** To optimize $L^{\text{SAM}}(\boldsymbol{w})$, we first have to find the worst-case perturbations $\boldsymbol{\epsilon}_s^*$ for the max problem. Foret *et al.* (Foret et al., 2020) practically approximate Eqn. (1) via the first-order expansion:

$$\boldsymbol{\epsilon}_s^* \approx \arg\max_{\|\boldsymbol{\epsilon}_s\|_2 \leq \rho} \boldsymbol{\epsilon}_s^\top \nabla_{\boldsymbol{w}} L(\boldsymbol{w}) = \rho \frac{\nabla_{\boldsymbol{w}} L(\boldsymbol{w})}{\|\nabla_{\boldsymbol{w}} L(\boldsymbol{w})\|_2}. \tag{7}$$

Then the gradient at the perturbed weight $\boldsymbol{w} + \boldsymbol{\epsilon}_s^*$ is computed for updating the model:

$$\nabla L^{\text{SAM}}(\boldsymbol{w}) \approx \nabla L(\boldsymbol{w})|_{\boldsymbol{w} + \boldsymbol{\epsilon}_s^*}. \tag{8}$$

Due to the two sequential gradient calculations involved for each iteration, the training speed of SAM is $2\times$ that of regular SGD training.

| Method | Perturbation Radius | Gradient Information | Computation | Time | Accuracy (%) |
|--------|--------------------|--------------------|-------------|------|--------------|
| AWP    | Small              | Yes                | 2×          | 2×   | 77.15        |
| RWP    | Medium             | No                 | 1×          | 1×   | 76.77        |
| ARWP   | Medium             | Yes                | 1×          | 1×   | 77.02        |
| m-RWP  | Large              | No                 | 2×          | 1×   | 77.82        |
| m-ARWP | Large              | Yes                | 2×          | 1×   | 78.04        |

Table 1: An overall comparison of different methods. RWP and ARWP utilize much larger perturbation radius than AWP, while m-RWP and m-ARWP can utilize even larger perturbation radius due to the improved convergence. The "gradient information" denotes whether utilizing gradient information for generating weight perturbation. RWP and ARWP requires only half of computation budget than AWP, while m-RWP and m-ARWP are able to paralleling the computation of the two gradient steps to half the training time. For performance comparison on ImageNet with ResNet-50, RWP and variants can achieve much more efficient generalization improvement than AWP.

**Random Weight Perturbation (RWP).** For optimizing $L^{\text{Bayes}}(\boldsymbol{w})$, we similarly sample a random perturbation $\boldsymbol{\epsilon}_r$ from a given distribution for each iteration and calculate the gradient at the perturbed weight $\boldsymbol{w} + \boldsymbol{\epsilon}_r$ for updating the model:

$$\nabla L^{\text{Bayes}}(\boldsymbol{w}) \approx \nabla L(\boldsymbol{w})|_{\boldsymbol{w}+\boldsymbol{\epsilon}_r}. \tag{9}$$

Note that the selection of the distribution plays an crucial role in the effectiveness of RWP. For modern DNNs, the loss function does not change with parameter scaling when ReLU-nonlinearities and batch normalization (Ioffe & Szegedy, 2015) are applied. Hence, it is essential to consider the filter-wise structure. Following the approach in (Bisla et al., 2022), we practically generate the RWP from a filter-wise Gaussian distribution, i.e., $\boldsymbol{\epsilon}_r \sim \mathbb{N}\left(\mathbf{0}, \sigma^2 diag(\{\|\boldsymbol{w}^{(j)}\|^2\}_{j=1}^{k})\right)$, with $\sigma$ controlling the perturbation magnitude.

**RWP requires much larger magnitude.** As the precise gradient direction is known, AWP is much more "effective" at perturbing the model compared to RWP. Consequently, in order to achieve a similar level of perturbation strength, the magnitude of RWP needs to be considerably larger than that of AWP. We carry out comparative experiments using a model $\boldsymbol{w}^*$ that has been well-trained with SGD and apply different perturbation magnitudes for both RWP and AWP. As dipicted in Figure 1, to attain a similar expected perturbed training loss $\mathbb{E}\left[L(\boldsymbol{w}^* + \boldsymbol{\epsilon})\right]$ (we calculate the mean perturbed loss over the entire training set with a batch size of 256), the perturbation radius of RWP would need to be roughly two orders of magnitude larger than that of AWP. Such large perturbations can introduce instability in training and cause convergence issues that degrade the performance.

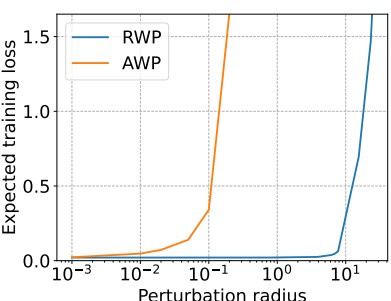

Figure 1: Expected training loss $\mathbb{E}[L(\boldsymbol{w}^* + \boldsymbol{\epsilon})]$ under different perturbation radii $\|\boldsymbol{\epsilon}\|_2$. The experiments employ a well-trained model $\boldsymbol{w}^*$ using SGD on CIFAR-10 with ResNet-18. Note that the x-axis is in logarithmic coordinates.

## 5 Improving Random Weight Perturbation

In this section, we propose to improve the performance of random weight perturbation from two primary perspectives. Firstly, we integrate the original loss objective to facilitate the trade-off between generalization and convergence. This enables enhanced convergence and meanwhile allows for larger perturbation magnitudes with improved generalization performance. Secondly, we focus on refining weight perturbation generation by incorporating historical gradient information. This enables a more adaptive and effective perturbation generation process.

### 5.1 Trade-off between Generalization and Convergence

In this subsection, we showcase that there exists a trade-off between convergence and generalization for RWP. We first analyze the smoothness properties of the expected loss function under RWP.

**Theorem 1 (Smoothness of RWP, Bisla et al. (2022))** *With Assumption 2 and 3, the function* $L^{\text{Bayes}}(\boldsymbol{w})$ *defined in Eqn. 2) is* $\min\{\frac{\alpha}{\sigma}, \beta\}$*-smooth.*

The above theorem indicates that a larger variance $\sigma$ for RWP will provide a better guarantee for smoothness, consequently leading to a smaller generalization error (Bisla et al., 2022). A minor observation is that a small magnitude of perturbation may not effectively contribute to the enhancement of smoothness when $\sigma$ is small, as in such cases $\frac{\alpha}{\sigma} > \beta$.

We then investigate the convergence properties of the following SGD optimization under RWP:

$$\boldsymbol{w}_{t+1} = \boldsymbol{w}_t - \gamma_t \nabla L_{\mathcal{B}}(\boldsymbol{w}_t + \boldsymbol{\epsilon}_r), \tag{10}$$

where $\mathcal{B}$ denotes the mini-batch data.

**Theorem 2 (Convergence of RWP in non-convex setting)** *Assume Assumption 1 and 3 hold. Let* $\boldsymbol{\epsilon}_r \sim \mathcal{N}(0, \sigma^2 I_{d \times d})$ *be the random perturbation and b be the batch size. Consider the sequence* $(\boldsymbol{w}_t)_{t \in \mathbb{N}}$ *generated by Eqn. (10), with a stepsize satisfying* $\gamma_t = \frac{\gamma_0}{\sqrt{t}}$ *and* $\gamma < \frac{1}{\beta}$*. Then we have*

$$\frac{1}{T} \sum_{t=1}^{T} \mathbb{E}\|\nabla L(\boldsymbol{w}_t)\|^2 \leq \frac{2\left(\mathbb{E}[L(\boldsymbol{w}_0)] - \mathbb{E}[L(\boldsymbol{w}^*)]\right)}{\gamma_0 \sqrt{T}} + (2\beta M + \beta^3 \sigma^2 d)\gamma_0 \frac{\log T}{\sqrt{T}} + 2\beta^2 \sigma^2 d. \tag{11}$$

We provide the proof in Appendix A and make several remarks: (1) Different from prior theoretical analysis (Andriushchenko & Flammarion, 2022; Mi et al., 2022) in SAM that assumes a decreasing magnitude on perturbation strength for achieving good convergence properties (e.g. $\rho_t = \frac{\rho_0}{\sqrt{t}}$ in Andriushchenko & Flammarion (2022)), we consider a more realistic setting with a non-decreasing $\sigma$ that is adopted in practice. (2) There are three terms that upper bound the expected training loss. The first two terms decrease at a rate of $\mathcal{O}(\frac{1}{\sqrt{t}})$ and $\mathcal{O}(\frac{\log(t)}{\sqrt{t}})$, respectively, which is consistent with the convergence rate of regular SGD. $\beta^3 \sigma^2 d$ is the additional variance term introduced by random weight perturbation. It can remain large when the perturbation radius ($\|\boldsymbol{\epsilon}_r\|$) is large, of which the expected radius is $\mathbb{E}[\|\boldsymbol{\epsilon}_r\|^2] = \sigma^2 d$, thereby slowing down convergence. The third term is a positive constant proportional to the perturbation variance $\sigma^2$, which prevents the effective reduction of the gradient norm beyond a certain point.

**A trade-off.** Based on the above two aspects, we can conclude that there exists a trade-off between generalization and convergence: for the sake of good smoothness properties and smaller generalization error, we need to increase the variance $\sigma^2$ for random weight perturbation. However, this can significantly increase the magnitude of the perturbation, and on the contrary, pose a convergence challenge.

### 5.2 Improving the Trade-off by Incorporating Original Loss

From the analysis in the last subsection, we know that there exists a trade-off between generalization and convergence in RWP. To achieve good generalization ability brought by large perturbation variance while enjoying a good convergence property, we propose to combine the original loss with the expected Bayes loss to improve the convergence of RWP, resulting in our **mixed-RWP** (m-RWP) loss objective:

$$L^{\text{m}}(\boldsymbol{w}) = \lambda L^{\text{Bayes}}(\boldsymbol{w}) + (1 - \lambda)L(\boldsymbol{w}), \tag{12}$$

where $\lambda \in [0, 1]$ is a pre-given balance coefficient. The two loss terms in our m-RWP objective are complementary to each other: the first $L^{\text{Bayes}}(\boldsymbol{w})$ provides a smoothed landscape that biases the network towards flat region, while the second $L(\boldsymbol{w})$ helps recover the necessary local information and better locates the minima that contributes to high performance. These two together could provide a both "local" and "global" viewing of the landscape — by optimizing $L^{\text{m}}(\boldsymbol{w})$, a good solution can be expected.

In practical implementation, we adopt a particular procedure for optimizing the objective function $L^{\text{Bayes}}(\boldsymbol{w})$ using RWP, i.e., in each training iteration, we sample one single random weight perturbation vector denoted as $\boldsymbol{\epsilon}_r$. To enhance the optimization process, we utilize two distinct batches of data, namely $\mathcal{B}_1$ and $\mathcal{B}_2$, for the two gradient steps involved. Note that this effectively enlarges the virtual batch size by a factor of two. The optimization process of our m-RWP can be formulated as follows:

$$\boldsymbol{w}_{t+1} = \boldsymbol{w}_t - \gamma_t \left[ \lambda \nabla L_{\mathcal{B}_1}(\boldsymbol{w}_t + \boldsymbol{\epsilon}_r) + (1 - \lambda) \nabla L_{\mathcal{B}_2}(\boldsymbol{w}_t) \right]. \tag{13}$$

In the subsequent analysis, we examine the smoothness and convergence properties of m-RWP. The proofs for these properties can be found in Appendix B and C, respectively.

**Theorem 3 (Smoothness of m-RWP)** *With Assumption 2 and 3, the function $L^{\text{m}}(\boldsymbol{w})$ defined in Eqn. (12) is $\min\{\frac{\lambda\alpha}{\sigma} + (1 - \lambda)\beta, \beta\}$-smooth.*

**Theorem 4 (Convergence of m-RWP in non-convex setting)** *Assume Assumption 1 and 3 hold. Let $\boldsymbol{\epsilon}_r \sim \mathcal{N}(0, \sigma^2 I_{d \times d})$ be the random perturbation and b be the batch size. Consider the sequence $(\boldsymbol{w}_t)_{t \in \mathbb{N}}$ generated by Eqn. (13), with a stepsize satisfying $\gamma_t = \frac{\gamma_0}{\sqrt{t}}$ and $\gamma < \frac{1}{\beta}$. Then we have*

$$\frac{1}{T} \sum_{t=1}^{T} \mathbb{E}\|\nabla L(\boldsymbol{w}_t)\|^2 \leq \frac{2 \left( \mathbb{E}[L(\boldsymbol{w}_0)] - \mathbb{E}[L(\boldsymbol{w}^*)] \right)}{\gamma_0 \sqrt{T}} + \left[ 2\beta M (2\lambda^2 - 2\lambda + 1) + \beta^3 \lambda^2 \sigma^2 d \right] \gamma_0 \frac{\log T}{\sqrt{T}} + 2\beta^2 \lambda^2 \sigma^2 d.$$

**Improved convergence.** Compared to the convergence properties of RWP as stated in Theorem 2, m-RWP offers immediate improvements in two aspects. Firstly, it reduces the variance term introduced by random weight perturbation ($\beta^3 \lambda^2 d \gamma_0 \frac{\log T}{\sqrt{T}}$) and the positive constant term ($2\beta^2 \sigma^2 d$) by a factor of $\lambda^2$. Secondly, m-RWP enables the use of two different data batches to compute the two gradient steps. This effectively doubles the batch size for each iteration training and thus reduces the gradient variance term ($2\beta M \gamma_0 \frac{\log T}{\sqrt{T}}$) by a factor of $2\lambda^2 - 2\lambda + 1$. As a result, the convergence is significantly improved. The convergence of RWP and m-RWP is compared in Figure 2.

**Better trade-off between generalization and convergence.** With the significant enhancement in convergence achieved by m-RWP, it becomes feasible to employ a larger perturbation variance. This allows us to enjoy the benefits from the improved smoothness properties offered by the increased perturbation variance while maintaining good convergence, potentially resulting in a better trade-off between generalization and convergence. Specifically, for attaining a similar convergence condition, denoted as $\beta^2 \sigma^2 d$ in Eqn. (11), the perturbation variance employed by m-RWP can be increased to $\sigma' = \sigma/\lambda$. The subsequent lemma demonstrates that, under certain conditions, the mixed loss function could provide better guarantees of smoothness compared to the original Bayes objective under similar convergence properties.

**Lemma 1** *Assuming that $\alpha < \beta\sigma < 2\alpha$, with Assumption 2 and 3, we have the following conclusions: 1) $L^{\text{Bayes}}(\boldsymbol{w})$ with a perturbation variance $\sigma$ is smoother than $L(\boldsymbol{w})$ and 2) $L^{\text{m}}(\boldsymbol{w})$ with a balance coefficient $\lambda \in (\frac{\beta\sigma - \alpha}{\alpha}, 1)$ and a perturbation variance $\sigma/\lambda$ is smoother than $L^{\text{Bayes}}(\boldsymbol{w})$.*

Please refer to Appendix D for the proof. In practical scenarios where constants such as the Lipschitz constant $\alpha$ and smoothness constant $\beta$ are unknown, we thus further validate our findings empirically. In Figure 3, we observe that m-RWP indeed achieves a better trade-off between generalization and convergence. Moreover, it demonstrates significantly improved performance by utilizing larger perturbation variances.

**Efficient parallel training.** Both SAM and m-RWP indeed involve two gradient steps for each iteration: $\nabla L(\boldsymbol{w})$ and $\nabla L(\boldsymbol{w} + \boldsymbol{\epsilon})$. However, the key distinction lies in the separability of these steps. In m-RWP, they are separable, and can be calculated independently and in parallel, whereas, in SAM, they need to be computed sequentially. This parallel computing capability of m-RWP allows for a halving of the training time, making it a highly efficient approach for large-scale problems.

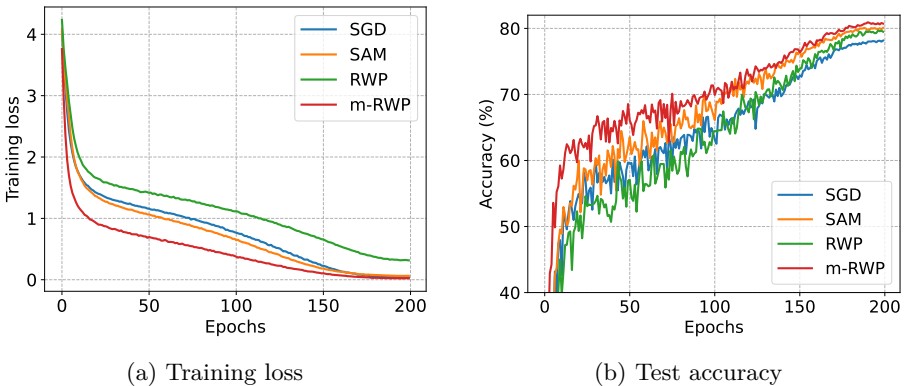

(a) Training loss

(b) Test accuracy

Figure 2: Training performance comparison of RWP and m-RWP. m-RWP significantly improve the convergence over RWP and leads to much better performance. The experiments are conducted on CIFAR-100 with ResNet-18. The perturbation variance $\sigma$ is set to 0.01.

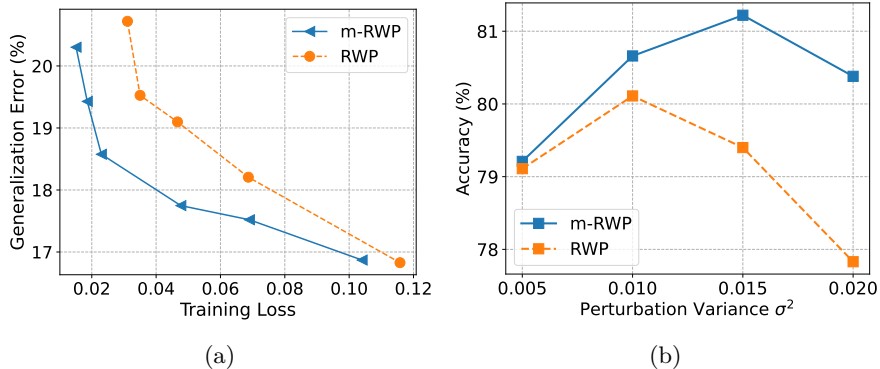

(a)

(b)

Figure 3: The mixed loss objective achieves a better trade-off between generalization and convergence. In (a), we conducted multiple runs of RWP and m-RWP with varying perturbation variances ($\sigma$) and recorded the final training losses and generalization errors (difference between training accuracy and test accuracy) for each trial. Our observations reveal that m-RWP achieves an improved trade-off between generalization error and convergence compared to RWP. In (b), m-RWP is capable of utilizing larger perturbation variances to achieve better generalization performance. The experiments are performed on CIFAR-100 with ResNet-18 and the balance coefficient $\lambda$ is set to 0.5.

## 5.3 Adaptive Random Weight Perturbation Generation

One limitation of previous methods for generating RWP is their exclusive reliance on weight norm, without utilizing any gradient information. This approach creates a disconnect between perturbation generation and the actual loss objective, resulting in overly "rough" perturbations.

Despite that for the efficiency considerations, we can not adopt an additional gradient step to assist the generation of RWP, the historical perturbed gradients are available and can be leveraged. We thus propose to utilize the historical gradient to better guide the generation of RWP. To achieve this, we employ an element-wise scaling of the gradient, taking into account the historical running sum of squares in each dimension. It is important to note that this technique is widely used in adaptive gradient descent optimizers such as Adam (Kingma & Ba, 2015) and RMSprop (Tieleman et al., 2012). The generation form of our

proposed method, called Adaptive Random Weight Perturbation (ARWP), is as follows:

$$\epsilon_{r,t} \sim \mathbb{N}\left(\mathbf{0}, \frac{\sigma^2 \boldsymbol{I}}{\sqrt{1 + \eta \sum_{i=1}^{t-1} \beta^{t-i-1} \boldsymbol{g}_i^2}}\right), \tag{14}$$

where $\eta$ and $\beta$ are two hyper-parameters. The rationale behind our generation approach is to move away from assigning a uniform perturbation magnitude ($\sigma$) to all parameters. Instead, we aim to introduce an adaptive perturbation strength based on the historical perturbed gradients. If the historical gradients are large, we want to assign a smaller perturbation magnitude, and vice versa. By incorporating such adaptability, we can tailor the perturbation magnitudes to the specific characteristics of each parameter. This allows us to account for variations in sensitivity and importance across different dimensions, leading to a more adaptive and effective generation of random perturbations. In practice, the perturbation variance $\sigma$ is progressively increasing (i.e., a cosine increasing strategy in Section 6.1 ) during the training of RWP/ARWP for gradually recovering the flat minima, as suggested by Bisla et al. (2022). Therefore, the primary enhancement of our ARWP lies in its adaptive generation of perturbations, rather than annealing the variance. Regarding convergence, we note that the variance of ARWP in Eqn. (14) is upper bounded by the corresponding variance of RWP in Eqn. (2). Therefore, the convergence properties of RWP in Theorem 2 and 4 still hold for ARWP.

Then combing with the filter-wise perturbation generation techniques from Bisla et al. (2022), our generation approach can be expressed as follows:

$$\epsilon_{r,t} \sim \mathbb{N}\left(\mathbf{0}, \frac{\sigma^2 diag(\{\|\boldsymbol{w}_t^{(j)}\|^2\}_{j=1}^k)}{\sqrt{1 + \eta \sum_{i=1}^{t-1} \beta^{t-i-1} diag(\{\|\boldsymbol{g}_i^{(j)}\|^2\}_{j=1}^k)}}\right). \tag{15}$$

Then combining the mixed loss objective in Eqn. (12), we propose our m-ARWP approach, which takes the same computational cost as SAM while being capable of halving the training time through parallel computation of the two gradient steps.

## 6 Experiments

In this section, we present extensive experimental results to demonstrate the efficiency and effectiveness of our proposed methods. We begin by introducing the experimental setup and then evaluate the performance over three standard benchmark datasets: CIFAR-10, CIFAR-100 and ImageNet. We also conduct ablation studies on the hyper-parameters and visualize the loss landscape to provide further insights.

### 6.1 Results

**Datasets & Models.** We experiment over three benchmark image classification tasks: CIFAR-10, CIFAR-100 (Krizhevsky & Hinton, 2009), and ImageNet (Deng et al., 2009). For CIFAR, we apply standard random horizontal flipping, cropping, normalization, and Cutout augmentation (DeVries & Taylor, 2017) (except for ViT, for which we use RandAugment (Cubuk et al., 2020)). For ImageNet, we apply basic data preprocessing and augmentation following the public Pytorch example (Paszke et al., 2017). We evaluate across a variety of representative architectures, including VGG (Simonyan & Zisserman, 2014), ResNet (He et al., 2016), WideResNet (Zagoruyko & Komodakis, 2016), and ViT (Dosovitskiy et al., 2021).

**Training Settings.** We compare the performance of six training methods: SGD, SAM, RWP, ARWP, m-RWP, and m-ARWP. It is worth noting that SGD, RWP, and ARWP share the same computation ($1\times$), while SAM, m-RWP and m-ARWP require twice the computational resources ($2\times$). For CIFAR experiments, we set the training epochs to 200 with batch size 256, momentum 0.9, and weight decay 0.001 (Du et al., 2022a; Zhao et al., 2022a), keeping the same among all methods for a fair comparison (except for ViT, we adopt a longer training schedule and provide the details in Appendix E). For SAM, we conduct a grid search for $\rho$ over $\{0.005, 0.01, 0.02, 0.05, 0.1, 0.2, 0.5\}$ and find that for CIFAR-10, $\rho = 0.05$ gives the best results for VGG16-BN, $\rho = 0.1$ suits best for ResNet-18 and WRN-28-10, while for CIFAR-100, $\rho = 0.1$ works best for VGG-16BN and $\rho = 0.2$ is optimal for ResNet-18 and WRN-28-10, which also coincides

Table 2: Results on CIFAR-10/100. We set the computation (FLOPs) and training time of SGD as $1\times$. The best accuracy is in bold and the second best is underlined.

| Model | Method | CIFAR-10 | CIFAR-100 | FLOPs | Time |
|---|---|---|---|---|---|
| VGG16-BN | SGD | $94.96_{\pm 0.15}$ | $75.43_{\pm 0.29}$ | $1\times$ | $1\times$ |
| | SAM | $95.43_{\pm 0.11}$ | $76.74_{\pm 0.22}$ | $2\times$ | $2\times$ |
| | RWP | $94.97_{\pm 0.07}$ | $76.56_{\pm 0.10}$ | $1\times$ | $1\times$ |
| | ARWP | $95.07_{\pm 0.23}$ | $77.07_{\pm 0.21}$ | $1\times$ | $1\times$ |
| | m-RWP | $\underline{95.49}_{\pm 0.16}$ | $\underline{77.68}_{\pm 0.14}$ | $2\times$ | $1\times$ |
| | m-ARWP | $\mathbf{95.61}_{\pm 0.23}$ | $\mathbf{77.98}_{\pm 0.19}$ | $2\times$ | $1\times$ |
| ResNet-18 | SGD | $96.10_{\pm 0.08}$ | $78.10_{\pm 0.39}$ | $1\times$ | $1\times$ |
| | SAM | $\underline{96.56}_{\pm 0.11}$ | $80.48_{\pm 0.25}$ | $2\times$ | $2\times$ |
| | RWP | $96.01_{\pm 0.31}$ | $80.21_{\pm 0.14}$ | $1\times$ | $1\times$ |
| | ARWP | $96.30_{\pm 0.03}$ | $80.71_{\pm 0.24}$ | $1\times$ | $1\times$ |
| | m-RWP | $96.58_{\pm 0.09}$ | $\underline{81.18}_{\pm 0.09}$ | $2\times$ | $1\times$ |
| | m-ARWP | $\mathbf{96.68}_{\pm 0.13}$ | $\mathbf{81.38}_{\pm 0.12}$ | $2\times$ | $1\times$ |
| WRN-28-10 | SGD | $96.85_{\pm 0.05}$ | $82.51_{\pm 0.24}$ | $1\times$ | $1\times$ |
| | SAM | $\mathbf{97.35}_{\pm 0.04}$ | $\mathbf{84.68}_{\pm 0.21}$ | $2\times$ | $2\times$ |
| | RWP | $96.73_{\pm 0.12}$ | $83.67_{\pm 0.14}$ | $1\times$ | $1\times$ |
| | ARWP | $96.89_{\pm 0.11}$ | $83.96_{\pm 0.09}$ | $1\times$ | $1\times$ |
| | m-RWP | $97.21_{\pm 0.04}$ | $84.37_{\pm 0.11}$ | $2\times$ | $1\times$ |
| | m-ARWP | $\underline{97.27}_{\pm 0.09}$ | $\underline{84.62}_{\pm 0.15}$ | $2\times$ | $1\times$ |
| ViT-S | Adam | $86.60_{\pm 0.03}$ | $63.66_{\pm 0.28}$ | $1\times$ | $1\times$ |
| | SAM | $87.48_{\pm 0.28}$ | $64.83_{\pm 0.24}$ | $2\times$ | $2\times$ |
| | RWP | $86.53_{\pm 0.04}$ | $63.67_{\pm 0.41}$ | $1\times$ | $1\times$ |
| | ARWP | $86.88_{\pm 0.09}$ | $64.12_{\pm 0.22}$ | $1\times$ | $1\times$ |
| | m-RWP | $\underline{87.71}_{\pm 0.13}$ | $\underline{65.67}_{\pm 0.09}$ | $2\times$ | $1\times$ |
| | m-ARWP | $\mathbf{88.18}_{\pm 0.19}$ | $\mathbf{66.13}_{\pm 0.13}$ | $2\times$ | $1\times$ |

with the hyper-parameters reported by Mi et al. (2022); Li & Giannakis (2023). For RWP and ARWP, we search $\sigma$ over $\{0.005, 0.01, 0.015, 0.02\}$ and set $\sigma = 0.01$ for both CIFAR-10/100 as it gives the optimal performance. We also adopt a cosine increasing schedule for $\sigma$ as described in (Bisla et al., 2022). For m-RWP and m-ARWP, we use $\sigma = 0.015$ and $\lambda = 0.5$ as it gives moderately good performance across different models. We set $\eta = 0.1$ and $\beta = 0.99$ as default choice. For ImageNet experiments, we set the training epochs to 90 with batch size 256, weight decay 0.0001, and momentum 0.9. We use $\rho = 0.05$ for SAM which aligns with (Foret et al., 2020; Kwon et al., 2021), $\sigma = 0.003$ for RWP and ARWP, and $\sigma = 0.005$, $\lambda = 0.5$ for m-ARWP. We employ $m$-sharpness with $m = 128$ for SAM as in (Foret et al., 2020; Kwon et al., 2021). For all experiments, we adopt cosine learning rate decay (Loshchilov & Hutter, 2016) with an initial learning rate of 0.1 and record the final model performance on the test set. Mean and standard deviation are calculated over three independent trials.

**CIFAR.** We begin by focusing on the CIFAR-10 and CIFAR-100 datasets. We evaluate the final test accuracy, total computation (in FLOPs), and training time for different methods. Detailed comparisons are presented in Table 2. We observe that ARWP consistently improves the performance RWP by 0.1-0.3% on CIFAR-10 and 0.3-0.5% on CIFAR-100, and m-ARWP consistently outperforms ARWP, improving performance by 1.6% on CIFAR-10 and 3.1% on CIFAR-100, confirming the effectiveness of our improvements on enhancing generalization. It is also worth mentioning that ARWP can achieve competitive, and sometimes even better, generalization performance compared to SAM, particularly with VGG16-BN (+0.3%) and ResNet-18 (+0.5%) models on CIFAR-100. Remarkably, ARWP achieves this with only half of the computation required by SAM. Additionally, m-ARWP outperforms SAM in most cases on CIFAR-10, and achieves improvements ranging from 0.1% to 1.3% on CIFAR-100. We can also observe that the mixing strategy alone contributes the most to performance improvement, e.g. with a notable increase of +0.97 % on CIFAR-100 with ResNet-18, as it entails a doubled computational overhead. In contrast, the adaptive perturbation strategy contributes an increase of +0.50%. Lastly, it is important to note that although both SAM and m-ARWP require twice the

Table 3: Results on ImageNet. We set the computation (FLOPs) and training time of SGD as 1×. The best accuracy is in bold and the second best is underlined.

| Model | Method | Accuracy (%) | FLOPs | Time |
|-------|--------|--------------|-------|------|
| VGG16-BN | SGD | 73.11 | 1× | 1× |
| | SAM | 74.65 | 2× | 2× |
| | RWP | 74.01 | 1× | 1× |
| | ARWP | 74.28 | 1× | 1× |
| | m-RWP | 75.45 | 2× | 1× |
| | m-ARWP | **75.79** | 2× | 1× |
| ResNet-18 | SGD | 70.32 | 1× | 1× |
| | SAM | 70.77 | 2× | 2× |
| | RWP | 70.46 | 1× | 1× |
| | ARWP | 70.71 | 1× | 1× |
| | m-RWP | 71.42 | 2× | 1× |
| | m-ARWP | **71.58** | 2× | 1× |
| ResNet-50 | SGD | 76.62 | 1× | 1× |
| | SAM | 77.15 | 2× | 2× |
| | RWP | 76.77 | 1× | 1× |
| | ARWP | 77.02 | 1× | 1× |
| | m-RWP | 77.82 | 2× | 1× |
| | m-ARWP | **78.04** | 2× | 1× |
| ViT-S/32 | AdamW | 68.12 | 1× | 1× |
| | SAM | 68.98 | 2× | 2× |
| | RWP | 68.40 | 1× | 1× |
| | ARWP | 68.74 | 1× | 1× |
| | m-RWP | 69.42 | 2× | 1× |
| | m-ARWP | **69.76** | 2× | 1× |

computation of SGD, the training time of m-RWP can be reduced by half compared to SAM through parallel computing. This reduction in training time is a valuable advantage of m-ARWP.

**ImageNet.** Next, we evaluate our proposed methods on ImageNet dataset, which has a substantially larger scale than CIFAR. We evaluate over four different architectures, namely VGG16-BN, ResNet-18, ResNet-50, and ViT-S/32, and present the results in Table 3. We observe that ARWP can achieve very competitive performance against SAM. For instance, with ResNet-18, ARWP achieves an accuracy of 70.71% compared to SAM's 70.77%, while with ResNet-50, ARWP achieves 77.02% accuracy compared to SAM's 77.15%. Notably, ARWP accomplishes this while requiring only half of the computational resources. Furthermore, m-ARWP significantly outperforms SAM, achieving 75.79% accuracy (+1.14%) with VGG16-BN, 71.58% accuracy (+0.81%) with ResNet-18, 78.04% accuracy (+0.89%) with ResNet-50, and 69.76% accuracy (+0.78%) with ViT-S/32. We note that models trained on ImageNet are typically under-trained. Therefore, the improved convergence of m-ARWP offers even more significant advantages over SAM. Moreover, the capability of parallel computing in m-ARWP is especially advantageous as the 2× longer training time in SAM would be prohibitively slow for large-scale problems.

## 6.2 Ablation Study and Visualization

**Impact of different data batches.** We conducted a further investigation into the impact of using the same or different data batches for the two gradient steps in m-ARWP and SAM. The results are presented in Table 4, and we made the following observations. For m-ARWP, both choices of using the same or different data batches yield comparable performance, with the use of different

Table 4: Effects of same/different data batches for two gradient steps. The experiments are conducted on CIFAR-100 with ResNet-18.

| Training | Same | Different |
|----------|------|-----------|
| SAM | 80.48±0.25 | 78.30±0.11 (↓ 2.18) |
| m-ARWP | 81.14±0.10 | 81.38±0.12 (↑ 0.24) |

batches resulting in a slightly better performance. On the other hand, for SAM, the two approaches yield starkly different results. Adopting different data batches for the adversarial attack and gradient propagation in SAM would undesirably degrade its generalization performance to that of plain SGD. This finding suggests that applying the same data batch for adversarial attack and gradient propagation is a crucial aspect for SAM's generalization improvement. We attribute this difference to the unique characteristic of AWP in SAM, which is specific to a particular batch of data. The perturbation computed over one batch in SAM can degenerate into meaningless noise when applied to another batch. In contrast, the perturbation used in m-ARWP is not associated with specific data instances, allowing for the use of different data batches to accelerate convergence with better efficiency.

**Sensitivity of hyper-parameters.**  There are four hyper-parameters involved in our approaches, $\eta$ and $\beta$ in ARWP, and additionally $\lambda$ and $\sigma$ in m-ARWP. We note that for the first three hyper-parameters, we use default values as $\eta = 1$, $\beta = 0.99$, $\lambda = 0.5$ across different experiments and only $\sigma$ needs to be tuned, thus virtually having the same number of hyper-parameters as SAM and RWP. To better understand their effects on performance, we test the performance under different choices of values. Specifically, we test on CIFAR-10/100 datasets with ResNet-18, and vary $\eta$ in $\{0.01, 0.1, 1\}$, $\beta$ in $\{0.9, 0.99, 0.999\}$, $\sigma$ in $\{0.005, 0.01, 0.015, 0.02\}$ and $\lambda$ in $\{0.1, 0.3, 0.5, 0.7, 0.9\}$. The results are in Figure 4. We observe that $\eta = 0.1$, $\beta = 0.99$, $\sigma = 0.015$, and $\lambda = 0.5$ are a robust choice that achieves moderately good performance on both CIFAR-10 and CIFAR-100.

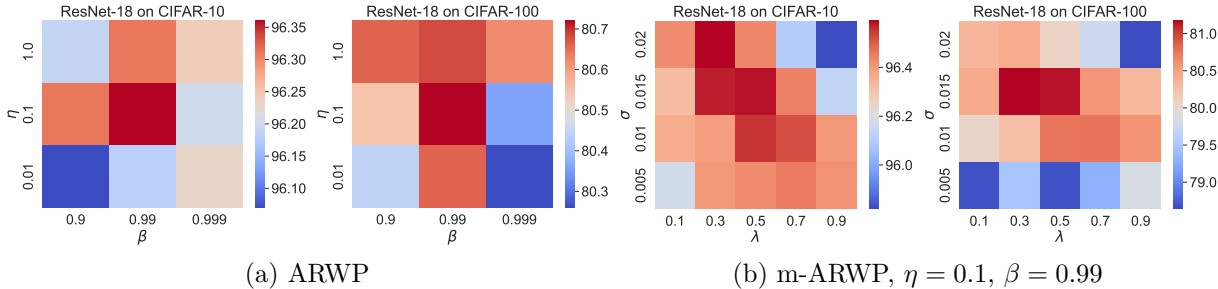

(a) ARWP                                       (b) m-ARWP, $\eta = 0.1$, $\beta = 0.99$

Figure 4: Performance under various hyper-parameter configurations.

**Loss landscape and Hessian spectrum.**  Finally, we compare the loss landscape and Hessian spectrum of SGD, RWP, ARWP, SAM, and m-ARWP. Following the plotting technique in (Li et al., 2018), we uniformly sample $50 \times 50$ grid points in the range of $[-1, 1]$ from random "filter-normalized" direction (Li et al., 2018), and for Hessian spectrum, we approximate it using the Lanczos algorithm (Ghorbani et al., 2019). In Figure 5, we observe that RWP, ARWP, SAM, and m-ARWP all achieve flatter loss landscape and smaller dominant eigenvalue ($\lambda_1$) compared to SGD. Additionally, ARWP demonstrates improved flatness and dominant eigenvalue compared to RWP. Moreover, m-ARWP exhibits even better flatness and a smaller dominant eigenvalue than SAM. Interestingly, from the perspective of the loss landscape, RWP and ARWP appear to have flatter landscapes compared to SAM and m-ARWP. However, they exhibit much larger Hessian dominant eigenvalues and worse generalization performance. This discrepancy may be attributed to the fact that the minima found by RWP and ARWP may not converge well and are even "over-smoothed" due to the large perturbation radius involved. Conversely, with the improved convergence of m-ARWP, we were able to reach a more precise minimum with better generalization performance.

# 7   Conclusion

In this work, we revisit the use of random weight perturbation for improving generalization performance. By uncovering the inherent trade-off between generalization and convergence in RWP, we propose a mixed loss objective that enables improved generalization while maintaining good convergence. We also introduce an adaptive strategy that utilizes historical gradient information for better perturbation generation. Extensive experiments on various architectures and tasks demonstrate that our improved RWP is able to achieve more efficient generalization improvement than AWP, especially on large-scale problems.

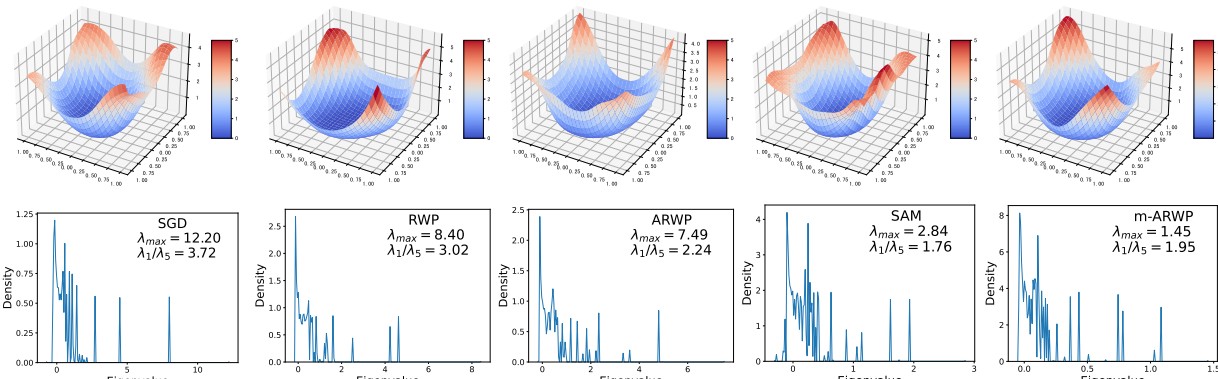

Figure 5: Loss landscape (**Up**) and the corresponding Hessian spectrum visualization (**Down**) of different methods. Models are trained on CIFAR-10 with ResNet-18.

### Acknowledgments

We thank the anonymous reviewers for their thoughtful comments that greatly improved the paper. This work was partially supported by National Key Research Development Project (2023YFF1104202), National Natural Science Foundation of China (62376155), Shanghai Municipal Science and Technology Research Program (22511105600) and Major Project (2021SHZDZX0102).

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

# A    Proof of Theorem 2

*Proof*    Denote $\boldsymbol{w}_{t+1/2} = \boldsymbol{w}_t + \boldsymbol{\epsilon}_{r,t}$. From Assumption 1, it follows that

$$L(\boldsymbol{w}_{t+1}) \leq L(\boldsymbol{w}_t) + \nabla L(\boldsymbol{w}_t)^\top (\boldsymbol{w}_{t+1} - \boldsymbol{w}_t) + \frac{\beta}{2}\|\boldsymbol{w}_{t+1} - \boldsymbol{w}_t\|^2 \tag{16}$$

$$=L(\boldsymbol{w}_t) - \gamma_t \nabla L(\boldsymbol{w}_t)^\top \nabla L_{\mathcal{B}}(\boldsymbol{w}_{t+1/2}) + \frac{\gamma_t^2 \beta}{2}\|\nabla L_{\mathcal{B}}(\boldsymbol{w}_{t+1/2})\|^2 \tag{17}$$

$$=L(\boldsymbol{w}_t) - \gamma_t \nabla L(\boldsymbol{w}_t)^\top \nabla L_{\mathcal{B}}(\boldsymbol{w}_{t+1/2})$$
$$+ \frac{\gamma_t^2 \beta}{2}\left(\|\nabla L_{\mathcal{B}}(\boldsymbol{w}_{t+1/2}) - \nabla L(\boldsymbol{w}_t)\|^2 - \|\nabla L(\boldsymbol{w}_t)\|^2 + 2\nabla L(\boldsymbol{w}_t)^\top \nabla L_{\mathcal{B}}(\boldsymbol{w}_{t+1/2})\right) \tag{18}$$

$$=L(\boldsymbol{w}_t) - \frac{\gamma_t^2 \beta}{2}\|\nabla L(\boldsymbol{w}_t)\|^2 + \frac{\gamma_t^2 \beta}{2}\|\nabla L_{\mathcal{B}}(\boldsymbol{w}_{t+1/2}) - \nabla L(\boldsymbol{w}_t)\|^2$$
$$- (1 - \beta\gamma_t)\gamma_t \nabla L(\boldsymbol{w}_t)^\top \nabla L_{\mathcal{B}}(\boldsymbol{w}_{t+1/2}) \tag{19}$$

$$\leq L(\boldsymbol{w}_t) - \frac{\gamma_t^2 \beta}{2}\|\nabla L(\boldsymbol{w}_t)\|^2 + \gamma_t^2 \beta\|\nabla L_{\mathcal{B}}(\boldsymbol{w}_{t+1/2}) - \nabla L(\boldsymbol{w}_{t+1/2})\|^2$$
$$+ \gamma_t^2 \beta\|\nabla L(\boldsymbol{w}_{t+1/2}) - \nabla L(\boldsymbol{w}_t)\|^2 - (1 - \beta\gamma_t)\gamma_t \nabla L(\boldsymbol{w}_t)^\top \nabla L_{\mathcal{B}}(\boldsymbol{w}_{t+1/2}). \tag{20}$$

The last step using the fact that $\|a - b\|^2 \leq 2\|a - c\|^2 + 2\|c - b\|^2$. Then taking the expectation on both sides gives:

$$\mathbb{E}[L(\boldsymbol{w}_{t+1})] \leq \mathbb{E}[L(\boldsymbol{w}_t)] - \frac{\gamma_t^2 \beta}{2}\|\nabla L(\boldsymbol{w}_t)\|^2 + \gamma_t^2 \beta M + \gamma_t^2 \beta^3 \sigma^2 d - (1 - \beta\gamma_t)\gamma_t \nabla L(\boldsymbol{w}_t)^\top \nabla L_{\mathcal{B}}(\boldsymbol{w}_{t+1/2}). \tag{21}$$

For the last term, we have:

$$
\begin{aligned}
\mathbb{E}[\nabla L(\boldsymbol{w}_t)^\top \nabla L_{\mathcal{B}}(\boldsymbol{w}_{t+1/2})] =& \mathbb{E}\left[\nabla L(\boldsymbol{w}_t)^\top \left(\nabla L_{\mathcal{B}}(\boldsymbol{w}_{t+1/2}) - \nabla L_{\mathcal{B}}(\boldsymbol{w}_t) + \nabla L_{\mathcal{B}}(\boldsymbol{w}_t)\right)\right] \\
=& \mathbb{E}\left[\|\nabla L(\boldsymbol{w}_t)\|^2\right] + \mathcal{C}.
\end{aligned}
\tag{22}
$$

where $\mathcal{C} = \mathbb{E}\left[\nabla L(\boldsymbol{w}_t)^\top \left(\nabla L_{\mathcal{B}}(\boldsymbol{w}_{t+1/2}) - \nabla L_{\mathcal{B}}(\boldsymbol{w}_t)\right)\right]$. Using the Cauchy-Schwarz inequality, we have

$$
\begin{aligned}
\mathcal{C} \leq& \mathbb{E}\left[\frac{1}{2}\|\nabla L(\boldsymbol{w}_t)\|^2 + \frac{1}{2}\|\nabla L(\boldsymbol{w}_{t+1/2}) - \nabla L(\boldsymbol{w}_t)\|^2\right] \\
\leq& \frac{1}{2}\mathbb{E}\|\nabla L(\boldsymbol{w}_t)\|^2 + \frac{\beta^2\sigma^2 d}{2}.
\end{aligned}
\tag{23}
$$

Plugging Eqn. (22) and (23) into Eqn. (21), we obtain:

$$
\begin{aligned}
\mathbb{E}[L(\boldsymbol{w}_{t+1})] \leq& \mathbb{E}[L(\boldsymbol{w}_t)] - \frac{\gamma_t^2 \beta}{2}\mathbb{E}\|\nabla L(\boldsymbol{w}_t)\|^2 + \gamma_t^2 \beta M + \gamma_t^2 \beta^3 \sigma^2 d - (1 - \beta\gamma_t)\gamma_t \mathbb{E}\left[\|\nabla L(\boldsymbol{w}_t)\|^2\right] \\
&+ (1 - \beta\gamma_t)\gamma_t \left(\frac{1}{2}\mathbb{E}\|\nabla L(\boldsymbol{w}_t)\|^2 + \frac{\beta^2\sigma^2 d}{2}\right)
\end{aligned}
\tag{24}
$$

$$
\leq \mathbb{E}[L(\boldsymbol{w}_t)] - \frac{\gamma_t}{2}\mathbb{E}\|\nabla L(\boldsymbol{w}_t)\|^2 + \gamma_t^2 \beta M + \frac{1}{2}(1 + \beta\gamma_t)\gamma_t \beta^2 \sigma^2 d
\tag{25}
$$

Taking summation over $T$ iterations, we have:

$$
\frac{\gamma_0}{2\sqrt{T}}\sum_{t=1}^T \mathbb{E}\|\nabla L(\boldsymbol{w}_t)\|^2 \leq \mathbb{E}[L(\boldsymbol{w}_0)] - \mathbb{E}[L(\boldsymbol{w}_T)] + (\beta M + \frac{1}{2}\beta^3\sigma^2 d)\gamma_0^2 \sum_{t=1}^T \frac{1}{t} + \frac{1}{2}\beta^2\sigma^2 d\gamma_0 \sum_{t=1}^T \frac{1}{\sqrt{t}}.
\tag{26}
$$

This gives:

$$
\frac{1}{T}\sum_{t=1}^T \mathbb{E}\|\nabla L(\boldsymbol{w}_t)\|^2 \leq \frac{2\left(\mathbb{E}[L(\boldsymbol{w}_0)] - \mathbb{E}[L(\boldsymbol{w}_T)]\right)}{\gamma_0\sqrt{T}} + (2\beta M + \beta^3\sigma^2 d)\gamma_0 \frac{\log T}{\sqrt{T}} + 2\beta^2\sigma^2 d
\tag{27}
$$

$$
\leq \frac{2\left(\mathbb{E}[L(\boldsymbol{w}_0)] - \mathbb{E}[L(\boldsymbol{w}^*)]\right)}{\gamma_0\sqrt{T}} + (2\beta M + \beta^3\sigma^2 d)\gamma_0 \frac{\log T}{\sqrt{T}} + 2\beta^2\sigma^2 d,
\tag{28}
$$

where $\boldsymbol{w}^*$ is the optimal solution. $\qquad\square$

## B  Proof of Theorem 3

*Proof*  For all $\boldsymbol{w}, \boldsymbol{v} \in \mathbb{R}^d$,

$$
\begin{aligned}
\|\nabla L^{\mathrm{m}}(\boldsymbol{w}) - \nabla L^{\mathrm{m}}(\boldsymbol{v})\| &\leq \lambda \left\|\nabla L^{\mathrm{Bayes}}(\boldsymbol{w}) - \nabla L^{\mathrm{Bayes}}(\boldsymbol{v})\right\| + (1 - \lambda)\left\|\nabla L(\boldsymbol{w}) - \nabla L(\boldsymbol{v})\right\| \\
&\leq \lambda \min\{\frac{\alpha}{\sigma}, \beta\}\|\boldsymbol{w} - \boldsymbol{v}\| + (1 - \lambda)\beta\|\boldsymbol{w} - \boldsymbol{v}\| \\
&= \min\{\lambda\frac{\alpha}{\sigma} + (1 - \lambda)\beta, \beta\}\|\boldsymbol{w} - \boldsymbol{v}\|.
\end{aligned}
$$

This shows that $L^{\mathrm{m}}(\boldsymbol{w})$ is $\min\{\frac{\lambda\alpha}{\sigma} + (1 - \lambda)\beta, \beta\}$-smooth. $\qquad\square$

## C  Proof of Theorem 4

*Proof*

$$
\begin{aligned}
L(\boldsymbol{w}_{t+1}) \leq& L(\boldsymbol{w}_t) + \nabla L(\boldsymbol{w}_t)^\top (\boldsymbol{w}_{t+1} - \boldsymbol{w}_t) + \frac{\beta}{2}\|\boldsymbol{w}_{t+1} - \boldsymbol{w}_t\|^2 \\
=& L(\boldsymbol{w}_t) - \gamma_t \nabla L(\boldsymbol{w}_t)^\top \left[\lambda\nabla L_{\mathcal{B}_1}(\boldsymbol{w}_{t+1/2}) + (1 - \lambda)\nabla L_{\mathcal{B}_2}(\boldsymbol{w}_t)\right]
\end{aligned}
\tag{29}
$$

$$
+ \frac{\gamma_t^2 \beta}{2}\|\lambda\nabla L_{\mathcal{B}_1}(\boldsymbol{w}_{t+1/2}) + (1 - \lambda)\nabla L_{\mathcal{B}_2}(\boldsymbol{w}_t)\|^2.
\tag{30}
$$

$$\mathbb{E}[L(\boldsymbol{w}_{t+1})] \leq \mathbb{E}[L(\boldsymbol{w}_t)] - \frac{\gamma_t^2 \beta}{2}\|\nabla L(\boldsymbol{w}_t)\|^2 + \gamma_t^2 \beta \left(\lambda^2 M + (1-\lambda)^2 M\right) + \gamma_t^2 \beta^3 \lambda^2 \sigma^2 d$$

$$- (1-\beta\gamma_t)\gamma_t\|\nabla L(\boldsymbol{w}_t)\|^2 + (1-\beta\gamma_t)\gamma_t(\frac{1}{2}\|\nabla L(\boldsymbol{w}_t)\|^2 + \frac{\gamma_t}{2}\lambda^2\beta^2\sigma^2 d) \tag{31}$$

$$= \mathbb{E}[L(\boldsymbol{w}_t)] - \frac{1}{2}\|\nabla L(\boldsymbol{w}_t)\|^2 + \gamma_t^2 \beta M \left(2\lambda^2 - 2\lambda + 1\right) + \frac{1}{2}(1+\beta\gamma_t)\gamma_t\lambda^2\beta^2\sigma^2 d \tag{32}$$

$$\tag{33}$$

Taking summation over $T$ iterations, we have:

$$\frac{\gamma_0}{2\sqrt{T}}\sum_{t=1}^{T}\mathbb{E}\|\nabla L(\boldsymbol{w}_t)\|^2 \leq \mathbb{E}[L(\boldsymbol{w}_0)] - \mathbb{E}[L(\boldsymbol{w}_T)] + \left(\beta M(2\lambda^2 - 2\lambda + 1) + \frac{1}{2}\beta^3\lambda^2\sigma^2 d\right)\gamma_0^2\sum_{t=1}^{T}\frac{1}{t}$$
$$+ \frac{1}{2}\beta^2\lambda^2\sigma^2 d\gamma_0\sum_{t=1}^{T}\frac{1}{\sqrt{t}}. \tag{34}$$

This gives:

$$\frac{1}{T}\sum_{t=1}^{T}\mathbb{E}\|\nabla L(\boldsymbol{w}_t)\|^2 \leq \frac{2\left(\mathbb{E}[L(\boldsymbol{w}_0)] - \mathbb{E}[L(\boldsymbol{w}^*)]\right)}{\gamma_0\sqrt{T}} + \left(2\beta M(2\lambda^2 - 2\lambda + 1) + \beta^3\lambda^2\sigma^2 d\right)\gamma_0\frac{\log T}{\sqrt{T}} + 2\beta^2\lambda^2\sigma^2 d.$$

$$\square$$

## D  Proof of Lemma 1

*Proof*   From Theorem 1 and 3 and the condition $\alpha < \beta\sigma < 2\alpha$, we can conclude that $L^{\mathrm{Bayes}}(\boldsymbol{w})|_\sigma$ is $\frac{\alpha}{\sigma}$-smooth since $\alpha < \beta\sigma$. Then leveraging $\lambda \in (\frac{\beta\sigma-\alpha}{\alpha}, 1)$, we obtain

$$\lambda > \frac{\beta\sigma - \alpha}{\alpha} \tag{35}$$

$$\Leftrightarrow \alpha(\lambda + 1) > \beta\sigma \tag{36}$$

$$\Leftrightarrow \alpha(\lambda^2 - 1) < (\lambda - 1)\beta\sigma \tag{37}$$

$$\Leftrightarrow \frac{\lambda^2\alpha}{\sigma} + (1-\lambda)\beta < \frac{\alpha}{\sigma}. \tag{38}$$

The Eqn. (37) is derived by multiplying both sides of the equation by $(\lambda - 1)$. Note that $L^{\mathrm{m}}(\boldsymbol{w})|_{\sigma/\lambda}$ is $\min\{\frac{\lambda^2\alpha}{\sigma} + (1-\lambda)\beta, \beta\}$-smooth. Thus, we can conclude that $L^{\mathrm{m}}(\boldsymbol{w})|_{\sigma/\lambda}$ is smoother than $L^{\mathrm{Bayes}}(\boldsymbol{w})|_\sigma$.  $\square$

## E  ViT training

To train the vision transformer model (Dosovitskiy et al., 2021), we adopt Adam Kingma & Ba (2015) as the base optimizer and train the models on CIFAR-10/100 datasets from scratch with Adam, SAM, RWP, and m-RWP. Specifically, we select the ViT-S model with input size $32 \times 32$, patch size 4, number of heads 8, and dropout rate 0.1. We train the models for 400 epochs with a batch size of 256, an initial learning rate of 0.0001, and a cosine learning rate schedule. For ARWP, we set $\sigma = 0.001$, and for m-ARWP, we set $\sigma = 0.002$ and $\alpha = 0.5$. For SAM, we perform a grid search for $\rho$ over $[0.001, 0.002, 0.005, 0.01, 0.05, 0.1, 0.2, 0.5]$ and finally select $\rho = 0.05$ for optimal. For ImageNet, we follow the implementation of Du et al. (2022b); Li & Giannakis (2023), where we train the model for 300 epochs with a batch size of 4096. The baseline optimizer is chosen as AdamW with weight decay 0.3. SAM relies on $\rho = 0.05$. We use $\sigma = 0.005$ for RWP/ARWP and $\sigma = 0.01$ for m-RWP/m-ARWP.

