# OpenReview forum: "Revisiting Random Weight Perturbation for Efficiently Improving Generalization"
_TMLR — Accepted by TMLR_

### Review · Reviewer_AyRx · 2024-01-22

**Summary Of Contributions:**

This paper proposes a new version of random weight perturbation (RWP) that mitigates the gap between adversarial weight perturbation (SAM) and RWP. They find that random weight perturbation require orders of larger perturbation sizes and propose to use other batches or historical gradient to help adjust the perturbation size for each parameter. Experiments show that the proposed algorithm can acheive comparable or even better performance without using as much compute as SAM.

**Audience:**

Yes

**Claims And Evidence:**

Yes

**Requested Changes:**

See weakness.

Minor:

p2 - lie two aspects -> lies in two aspects

**Strengths And Weaknesses:**

Strength:

- A clever idea to bring RWP and AWP by finding what’s the important factors of the perturbations (maybe it is only the scale)
- they justify the propose approach with a theoretical analysis, showing that perturbation size has a large effect on convergence
- the experiments are impressive, delivering good performance with much less compute

Weakness:

- Theoretical results. I am not an expert in optimization and I didn’t check the details of the theorems. Looking at the final theorem, I am wondering why in Thm 1, a larger peturbation always imply worse convergence. If so, why would we adopt pertuabtions at all. I believe that there should be some tradeoffs there, and it’s good to include it as a whole picture.
- It seems that the mixed training delivers most of the empirical improvement, and the improvement of the adaptive scheme ARWP seems to have marginal improvement, sometimes inferior to SAM (see Table 2). It is good to have an ablation on the contributions of the two techniques.
- No ViT result on ImageNet, which is strange (Table 2 has ViT results on CIFARs). It would be good to show how it works for ViTs as well, which is the  trending method now.

---

> ### Author Response · Authors · 2024-02-12
> **Response to Reviewer AyRx**
>
> Thank you for your valuable and insightful comments! In the following, we provide our point-by-point response and hope our response helps address your concerns. We also look forward to the subsequent discussion which may further help solve the current issues.
>
> **Q1: Why would we adopt pertuabtions at all; I believe that there should be some tradeoffs there, and it’s good to include it as a whole picture.**
>
> **A1:**: Thank you for your insightful question and suggestion. We appreciate your understanding of the tradeoff between generalization and convergence in the context of random weight perturbation (RWP). To achieve effective smoothing of the loss objective and improve generalization, it is necessary to introduce significant perturbations in RWP. However, these perturbations can present challenges to convergence, emphasizing the tradeoff between generalization and convergence.
>
> In response to your suggestion, we have made revisions to the paper. We have included a discussion on this tradeoff before introducing our methodology in Section 5.1. Additionally, we have provided a thorough smoothness analysis of the loss objective and discussed how our mixing strategy improve the tradeoff (Theorem 1, 3; Lemma 1). We hope that these additions help to clarify the necessity of perturbation and strengthen the connection between the tradeoff analysis and our proposed methodology.
>
>
> **Q2: It is good to have an ablation on the contributions of the two techniques.**
>
> **A2:** Thanks for your suggestion. In the revision, we have included the performance of RWP/ARWP/m-RWP/m-ARWP in our main results (Table 2/3) to showcase the contributes of the two techniques and add corresponding discussion on their respective contributions (Section 6.1).
>
> Analyzing the results for ResNet-18 in the following table, we observe that the mixing strategy alone contributes the most to performance improvement, e.g. with a notable increase of +0.97% on CIFAR-100. In contrast, the adaptive perturbation strategy contributes an increase of +0.50%. It is worth noting that the mixing strategy entails a doubled computational overhead.
> Moreover, we can observe that implementing the adaptive strategy on RWP alone brings about a more significant performance improvement than further applying the adaptive strategy to m-RWP (+0.50% v.s. +0.20% on CIFAR-100)
>
>
>
> ResNet-18 | RWP | ARWP | m-RWP | m-ARWP
> --- | --- | --- | --- | ---
> FLOPs| 1x | 1x | 2x | 2x
> CIFAR-10 | 96.01$_{\pm0.31}$ | 96.30$_{\pm0.03}$ | 96.58$_{\pm0.09}$ | 96.68$_{\pm0.13}$
> CIFAR-100 | 80.21$_{\pm0.14}$ | 80.71$_{\pm0.24}$ | 81.18$_{\pm0.09}$ | 81.38$_{\pm0.12}$
>
>
> **Q3: It would be good to show how it works for ViTs as well, which is the trending method now.**
>
> **A3:** Thank you for your suggestion. In the revised version, we have added ViT experiments in Table 3, following the settings of [1]. The following table confirms the significant advantages of our RWP approaches over SAM in terms of performance and training time.
>
> ViT-S/32 | AdamW | SAM | RWP | ARWP | m-RWP | m-ARWP
> ---|---|---|---|---|---|---
> FLOPs| 1x|2x|1x|1x|2x|2x
> Time| 1x| 2x| 1x| 1x|1x|1x
> Accuracy (%)|68.12|68.98| 68.40 | 68.74 | 69.42 | 69.76 |
>
> [1] Li, Bingcong, and Georgios B. Giannakis. "Enhancing sharpness-aware optimization through variance suppression." Advances in Neural Information Processing Systems (2023).

---

> > ### Comment · Reviewer_AyRx · 2024-02-13
> >
> > Thank the authors for providing a very detailed and helpful response and revision. The new theoretical justification indeed helps the readers to get the whole picture of using RWP. It's also good to see the proposed method also works for ViT on ImageNet with around 1.6% improvement in accuracy. The ablation experiments on CIFARs and ImageNet also backup the effectiveness of the two proposed approaches. Nevertheless, the adaptive technique seems less useful than the m-RWP (marginal improvements), while the main argument in the paper is that the deficiency of RWP is caused by the large perturbation. Maybe the story can also be adjusted a bit to accommodate deficiency from both sides.
> >
> > Overall, I acknowledge the authors' efforts in revising the manuscript. I now lean towards acceptance and I am happy to discuss more with the other reviewers.

---

> > > ### Author Response · Authors · 2024-02-13
> > > **Response to Reviewer AyRx**
> > >
> > > We sincerely thank you for your timely and positive feedback! We are delighted that our response and revision have helped address your concerns and get the whole picture of using RWP. As you pointed out, the primary improvement lies in m-RWP, while the adaptive technique offers less pronounced improvements but with minimal overhead. We will revise this part with less emphasis after receiving feedback from other reviewers. Many thanks for your insightful comments again!

---

### Review · Reviewer_HxDU · 2024-01-24

**Summary Of Contributions:**

The paper proposes a new distribution for sampling weight perturbations when training neural networks. While previous work just makes use of Gaussian distributions, this paper uses a mixture of a delta function placed on zero and a Gaussian. Moreover, they reuse the moment estimation techniques from adaptive optimisers to set the variance of the Gaussians dynamically.

**Audience:**

Yes

**Claims And Evidence:**

No

**Requested Changes:**

* The authors should provide a more detailed description of the hyperparameter tuning process, and should tune hyperparameters for all methods considered in the empirical evaluation.
* The authors should demonstrate the practical and statistical significance of their experimental findings.
* The paper should be revised to make it clear how the experiment in Section 4 is related to the new method. Alternatively, the authors could provide a more detailed discussion/demonstration of the tradeoff between convergence and generalisation of existing methods as the magnitude of perturbations are varied. A subsequent demonstration that the proposed perturbation distribution enables a different/better set of tradeoffs would greatly strengthen the paper.
* The paper claims that an advantage over previous theoretical analysis is that the result in this paper does not require the variance to be annealed throughout training. However, the proposed adaptive variance method will anneal the variance throughout training. Can the authors add some additional discussion to resolve the inconsistency?

**Strengths And Weaknesses:**

* I like the high-level organisation of the paper. Starting with identification of the drawbacks of existing approaches, followed by suggesting how they can be overcome is a useful way to frame new methodological developments.
* Unfortunately the instantiation of this organisation is not convincing. The analysis experiment presented by and in Figure 1 does not seem to directly influence the methodological development in the next section.
* The theoretical analysis does not add any new insights over previous work. Classic results on the convergence of SGD already take the variance of the gradient estimator into account, so it is already known that reducing the variance of gradient noise leads to better convergence. The proof of the theorems in this paper simply add the already known variance of the noise injected by RWP to the usual noise considered by typical SGD analysis.
* The argument used to support the proposed method is that, by sampling weight perturbations from a distribution with smaller variance, convergence can be improved. However, the convergence analysis suggests that the best convergence is achieved by not adding a perturbation at all. This is obviously at odds with the original motivation for using AWP and RWP, which is to improve generalisation rather than convergence. The theoretical and empirical analyses do not explore this tradeoff.
* The empirical results are unreliable, as the authors tuned hyperparameters for their own methods but not others. There are also few details of how the hyperparameters were systematically tuned. Moreover, even with the unfair empirical comparison it is not clear that the results are actually statistically or practically significant.

---

> ### Author Response · Authors · 2024-02-12
> **Response to Reviewer HxDU**
>
> Thank you for your valuable and detailed comments! In the following, we provide our point-by-point response and hope our response helps address your concerns. We also look forward to the subsequent discussion which may further help solve the current issues.
>
> **Q1: The analysis experiment presented by and in Figure 1 does not seem to directly influence the methodological development in the next section.**
>
> **A1:** Thank you for bringing this issue to our attention. To achieve effective smoothing of the loss objective and improve generalization, it is necessary to introduce significant perturbations in RWP, which can be much larger than AWP as illustrated in Figure 1.
> In the revised version, we have described such smoothing effects of RWP (Theorem 1), and further provided a description of the tradeoff between generalization and convergence for RWP to connect the methodology (Section 5.1). Specifically, we highlight the challenge of achieving effective generalization improvement requiring a significant perturbation variance, which can pose difficulties for convergence. We have included this discussion before delving into the methodology section in the revision. Furthermore, we have revised our goal from "improving convergence" to the more precise objective of "improving the tradeoff". We hope that these revisions establish a more natural connection between the analysis and methodology, enhancing the overall coherence of our research.
>
>
> **Q2: The theoretical analysis does not add any new insights over previous work.**
>
> **A2:** Our convergence analysis follows the standard framework of stochastic optimization and builds upon previous works on the convergence analysis of SAM (e.g. [1-5]). For the new insights, we note that we are the first to specifically analyze the convergence of SGD with random weight perturbations. This setting involves significantly larger perturbation radii compared to the adversarial perturbations in SAM. Moreover, we also provide a smoothness analysis of the mixed loss objective and show that it can achieve better smoothness guarantees than RWP under similar convergence guarantees.
>
>
> **Q3: Odds with the original motivation for using AWP and RWP, which is to improve generalisation rather than convergence. The theoretical and empirical analyses do not explore this tradeoff.**
>
> **A3:** Thanks. In the revised version, we have clarified our motivation by introducing the tradeoff between generalization and convergence: achieving effective generalization improvement requires a significant perturbation variance, which can pose difficulties for convergence.
> Following your suggestion, we have included this before our methodology that uses a mixing strategy to improve such trade-offs.  We have also provided a theoretical analysis (Lemma 1) that demonstrates how our mixed strategy could achieve an improved tradeoff between generalization and convergence under certain conditions. Furthermore, we have also conducted experiments to validate these findings, as depicted in Figure 3.
>
>
>
> **Q4: The empirical results are unreliable, as the authors tuned hyperparameters for their own methods but not others.**
>
> **A4:** Thanks for pointing out the issue. In the revised version, we have conducted a grid search over $\\{0.005, 0.01, 0.02, 0.05, 0.1, 0.2, 0.5\\}$ to determine the optimal value of $\rho$ for SAM, as you recommended. Consequently, some of SAM' results are strengthened accordingly, and the optimal $\rho$ aligns with recent works, such as [2,5]. Still, we observe significant improvement, particularly on large-scale datasets,  such as a notable increase of +0.89% on ImageNet with ResNet-50. Furthermore, our paper emphasizes the advantage of **efficient** generalziation improvement with RWP over SAM: RWP enables parallel computing of the two gradient steps and offers computationally lightweight perturbation, thereby providing a distinct advantage.

---

> ### Author Response · Authors · 2024-02-12
> **Response to Reviewer HxDU (Requested Changes)**
>
> **Q1: The authors should provide a more detailed description of the hyperparameter tuning process, and should tune hyperparameters for all methods considered in the empirical evaluation.**
>
> **A1:** Following your suggestion, we have performed a grid search to determine the optimal $\rho$ of SAM and subsequently revised the results accordingly in the revision. Our hyper-parameters are consistent with those reported in previous published works [2,5], ensuring comparability and maintaining consistency in our study.
>
> **Q2: The authors should demonstrate the practical and statistical significance of their experimental findings.**
>
> **A2:** Following your suggestion, we have revised the results of SAM by performing a grid search for $\rho$ across different models and datasets. We still observe significant improvements, such as a notable increase of +0.90% with ResNet-18 on CIFAR-100 under three independent trials. Notably, our improvements over SAM are even more significant on the large-scale ImageNet dataset, with improvements of +1.14% with VGG16-BN, +0.81% with ResNet-18, +0.89% with ResNet-50, and +0.78% with ViT-S/32.
>
> Furthermore, it is important to highlight another crucial aspect of the advantages of RWP. RWP enables parallel computing of the two gradient steps and offers computationally lightweight perturbations, leading to improved efficiency. This efficiency enhancement is a valuable characteristic of RWP, making it a promising approach for practical applications.
>
> **Q3: Should be revised to make it clear how the experiment in Section 4 is related to the new method; provide a more detailed discussion/demonstration of the tradeoff between convergence and generalisation as the magnitude of perturbations are varied; subsequent demonstration that enables a different/better set of tradeoffs would greatly strengthen the paper.**
>
> **A3:** Thanks for the detailed suggestions. In the revision, several key changes have been made: **1)** We have included a description highlighting the trade-off between generalization and convergence, underscoring the need for larger perturbation variances. This description is presented before introducing our methodology, providing a clear context for the reader; **2)** In Section 5.1, we have provided a detailed discussion on the trade-off between generalization and convergence, as the magnitude of perturbations are varied (Figure 3);
> **3)** Following your suggestion, we both theoretically and empirically show that our mixing strategy could potentially provide a better trade-off and lead to better generalization performance, please refer to Lemma 1 and Figure 3.
>
> **Q4: The proposed adaptive variance method will anneal the variance throughout training.**
>
> **A4:** Thanks for providing the new aspect. As you pointed out, our ARWP method incorporates an adaptive adjustment of the variance magnitude through the inverse scaling of the running average of historical squared gradients, i.e., Eqn. (14) in the paper:
> $$
> \boldsymbol{\epsilon}\_{r,t}^{\rm ARWP} \sim \mathbb{N} \left ( \mathbf{0}, \frac{\sigma^2 \boldsymbol{I}}{\sqrt{1+\eta \sum_{i=1}^{t-1} \beta^{t-i-1} \boldsymbol{g}_i^2}} \right ).
> $$
> We would like to clarify that no need to anneal the variance throughout training is not our advantage.
> In practice, we employ a **progressively increasing** strategy (e.g., a cosine increasing in Sec. 6.1)  for $\sigma$ during the training of RWP/ARWP, as suggested by [6], rather than annealing the variance. The rationale behind this strategy is that "the more you progress with the training, the more you care to recover flat regions in the loss landscape"  [6], so as to advocate better generalization performance.
>
> Regarding the scaling w.r.t. the historical gradients, this technique is widely used in adaptive gradient descent optimizers. By incorporating this approach, we can assign adaptive magnitudes in perturbation generation to adapt to different parameters and training stages of DNNs. Consequently, the primary improvement of ARWP lies in its adaptive generation of perturbations to enhance the effectiveness of promoting generalization, rather than relying on variance annealing. We have provided further clarification on this aspect in our revised version (see Section 5.3).

---

> > ### Author Response · Authors · 2024-02-12
> > **Response to Reviewer HxDU (References)**
> >
> > [1] Andriushchenko, Maksym, and Nicolas Flammarion. "Towards understanding sharpness-aware minimization." International Conference on Machine Learning (2022).
> >
> > [2] Mi, Peng, et al. "Make sharpness-aware minimization stronger: A sparsified perturbation approach." Advances in Neural Information Processing Systems (2022).
> >
> > [3] Mueller, Maximilian, et al. "Normalization Layers Are All That Sharpness-Aware Minimization Needs." Advances in Neural Information Processing Systems (2023).
> >
> > [4] Jiang, Weisen, et al. "An Adaptive Policy to Employ Sharpness-Aware Minimization." International Conference on Learning and Representation (2023).
> >
> > [5] Li, Bingcong, and Georgios B. Giannakis. "Enhancing sharpness-aware optimization through variance suppression." Advances in Neural Information Processing Systems (2023).
> >
> > [6] Bisla, Devansh, Jing Wang, and Anna Choromanska. "Low-pass filtering sgd for recovering flat optima in the deep learning optimization landscape." International Conference on Artificial Intelligence and Statistics (2022).

---

### Review · Reviewer_iFGJ · 2024-02-06

**Summary Of Contributions:**

This paper proposes an improvement of the random weight perturbation-based optimization algorithm. It has widely been known that optimizing the worst-case error around a parameter reduces the sharpness of the loss surface near the local optima, and leads to a better generalization. The most popular approach to implement this idea is SAM, where the adversarially chosen noises perturb the parameters. On the other hand, SAM has a loose connection with the Bayes objective based on the random weight perturbation (RWP), but the corresponding performance is not on par with SAM. This paper empirically and theoretically analyzes why the RWP may be ineffective, and proposes ways to fix it. Specifically, the mixed RWP is proposed, where the Bayes objective with the RWP is combined with the ordinary SGD objective. The theory shows that mixing can reduce the expected gradient norm, leading to a faster convergence. An additional contribution is to adaptively set the standard deviation of the Gaussian weight perturbation based on the empirical gradients, similar in spirit to the adaptive optimization methods. The proposed method, entitled m-ARWP, is demonstrated on various image classification benchmarks.

**Audience:**

Yes

**Broader Impact Concerns:**

I don't see any ethical issues or implications related to the contribution in this paper.

**Claims And Evidence:**

Yes

**Requested Changes:**

- The main experimental results do not include m-RWP. It would be good to have it as well to see the net effect of the mixing.
- Theorem 2 only guarantees improved convergence, but it does not say anything about the flatness. It would be good to have a related theory, but if not feasible, at least, the experiments to see the effect of the hyperparameter lambda on the flatness of the loss surface would be helpful; the current experiments only include the comparison to baselines with a fixed lambda value.
- While m-RWP is theoretically backed up, ARWP and m-ARWP are only empirically validated. Would it be hard to incorporate the modified variance term into the Theorem 2?

**Strengths And Weaknesses:**

- The paper is well-written and easy to follow.
- The proposed algorithm is simple and easy to implement, has some benefits over SAM (in terms of computational complexity), and yet outperforms SAM.
- The experiments are diverse, and done for large-scale architectures. The proposed method works well for every case. Ablation studies on the hyperparameters are also provided.

---

> ### Author Response · Authors · 2024-02-12
> **Response to Reviewer iFGJ**
>
> Thank you for your valuable and insightful comments! In the following, we provide our point-by-point response and hope our response helps address your concerns. We also look forward to the subsequent discussion which may further help solve the current issues.
>
> **Q1: The main experimental results do not include m-RWP. It would be good to have it as well to see the net effect of the mixing.**
>
> **A1:** Thanks for your suggestion. We have added the m-RWP results in Table 2 and 3 in the revision. Mixing indeed contributes the majority of the performance improvement.
> For example, analyzing the results for ResNet-18 in the following table, we observe that the mixing strategy alone contributes the most to performance improvement, e.g. with a notable increase of +0.97% on CIFAR-100. In contrast, the adaptive perturbation strategy contributes an increase of +0.50%.
>
> ResNet-18 | RWP | ARWP | m-RWP | m-ARWP
> --- | --- | --- | --- | ---
> FLOPs| 1x | 1x | 2x | 2x
> CIFAR-10 | 96.01$_{\pm0.31}$ | 96.30$_{\pm0.03}$ | 96.58$_{\pm0.09}$ | 96.68$_{\pm0.13}$
> CIFAR-100 | 80.21$_{\pm0.14}$ | 80.71$_{\pm0.24}$ | 81.18$_{\pm0.09}$ | 81.38$_{\pm0.12}$
>
>
> **Q2: Theory about flatness; The effect of the hyperparameter lambda on the flatness of the loss surface.**
>
> **A2:** Thanks for the suggestions. Using random weight perturbation is well shown to effectively improve the smoothness of the loss objective [1], thus leading to smaller generalization error between training and test data. We thus add smoothness analysis into our revision.
> **1)** We have analyzed the smoothness properties of RWP and m-RWP (Theorem 1 and 3), and shown that under certain assumptions, the m-RWP loss objective can be smoother than the RWP objective under similar convergence guarantees (Lemma 1), thus leading to better generalization performance.
> Furthermore, we have empirically validated these findings. Figure 3 showcases the results of our experiments, illustrating that m-RWP achieves a better trade-off between generalization error and convergence. This improved trade-off leads to enhanced performance, emphasizing the effectiveness and benefits of the m-RWP approach.
> **2)** Following your suggestion, we vary the $\lambda$ on CIFAR-100 using ResNet-18 with $\sigma=0.015$. We measured the generalization error, which indicates the model's ability to generalize to new data and adapt to distribution changes. Our observations reveal that when using pure RWP with $\lambda=1$, the model achieves the smallest generalization error. However, it also exhibits the worst training accuracy. Conversely, using a vanilla loss with $\lambda=0$ results in the best training accuracy but the largest generalization error. By incorporating the mixing strategy, we are able to achieve a better trade-off between training accuracy and generalization error. This leads to improved test performance, where the model strikes a balance between accurate training and robust generalization.
>
> $\lambda$ | $0$  | $0.25$ | $0.5$ | $0.75$ | $1.0$
> --- | --- | --- | --- | --- | --- |
> Training Acc. |**99.38**|99.34|98.86| 97.76| 94.31
> Test Acc. |78.01|81.12|**81.25**| 80.79| 79.43
> Generalization err. |21.37|18.22|17.61|16.97|**14.88**
>
>
> **Q3: Incorporate the modified variance term into the Theorem 2.**
>
> **A3:** Thanks for the suggestion. From the following generation formulation of RWP and ARWP, we can observe that the variance of ARWP is upper bounded by the corresponding variance of RWP. Therefore, the convergence properties established for  RWP still hold for ARWP. However, in order to obtain a tighter upper bound, knowledge of the lower bound of the gradient norm $\|\boldsymbol{g}\|$ is required, which is currently unknown. Furthermore, as the gradient norm $\|\boldsymbol{g}\|$ decreases during convergence, the difference between RWP and ARWP becomes smaller, and thus their convergence properties will be alike. To avoid redundancy of theorems, we omit the convergence theorem of ARWP but add a discussion on convergence in the revision (see Section 5.3).
>
> $$
> \boldsymbol{\epsilon}\_{r,t}^{\rm RWP} \sim \mathbb{N} \left ( \mathbf{0},{\sigma^2} \boldsymbol{I} \right ),  ~~~~\boldsymbol{\epsilon}\_{r,t}^{\rm ARWP} \sim \mathbb{N} \left ( \mathbf{0}, \frac{\sigma^2 \boldsymbol{I}}{\sqrt{1+\eta \sum_{i=1}^{t-1} \beta^{t-i-1} \boldsymbol{g}_i^2}} \right ).
> $$
>
> [1] Bisla, Devansh, Jing Wang, and Anna Choromanska. "Low-pass filtering sgd for recovering flat optima in the deep learning optimization landscape." International Conference on Artificial Intelligence and Statistics (2022).

---

### Decision · Action_Editor_BZ8U · 2024-03-12

**Recommendation:** Accept as is

**Comment:**

- The paper is well-written and easy to follow.
- The proposed algorithm is simple and easy to implement.
- The proposed algorithm has some benefits over SAM (in terms of computational complexity), and yet outperforms SAM.
- The experiments are diverse and done for large-scale architectures.
- The proposed method works well in all experiments.
- Ablation studies on the hyperparameters are also provided.

The authors were active during the rebuttal period.

**Audience:**

As stated by one of the reviewers:
"(...) there will be some people in the TMLR audience that find it interesting."

**Claims And Evidence:**

This paper proposes an improvement of the random weight perturbation-based optimization algorithm (RWP). Previous work makes use of Gaussian distributions for sampling weight perturbations when training neural networks, while this paper uses a mixture of a delta function placed on zero and a Gaussian. Moreover, this paper proposes to use other batches or historical gradients to help adjust the perturbation size for each parameter.

The reviewers raised some questions and during the rebuttal period, the authors worked hard to address all of them. One of the remaining issues is the following:
"The adaptive technique seems less useful than the m-RWP with marginal improvements, and the authors are further encouraged to revise the motivation part."

However, in my assessment, this is not critical for the paper.

All claims are clearly stated and enough evidence is provided, thus, the paper can be accepted in its current form.